# Sleep and Mood Disturbances during the COVID-19 Outbreak in an Urban Chinese Population in Hong Kong: A Longitudinal Study of the Second and Third Waves of the Outbreak

**DOI:** 10.3390/ijerph18168444

**Published:** 2021-08-10

**Authors:** Chun Sing Lam, Branda Yee-Man Yu, Denise Shuk Ting Cheung, Teris Cheung, Simon Ching Lam, Ka-Fai Chung, Fiona Yan-Yee Ho, Wing-Fai Yeung

**Affiliations:** 1Faculty of Medicine, School of Pharmacy, The Chinese University of Hong Kong, Hong Kong, China; jasonlamcs10@link.cuhk.edu.hk; 2Department of Psychology, The University of Hong Kong, Hong Kong, China; brandayu@connect.hku.hk; 3Li Ka Shing Faculty of Medicine, School of Nursing, The University of Hong Kong, Hong Kong, China; denisech@connect.hku.hk; 4School of Nursing, The Hong Kong Polytechnic University, Hong Kong, China; teris.cheung@polyu.edu.hk; 5School of Nursing, Tung Wah College, Hong Kong, China; simonlam@twc.edu.hk; 6Department of Psychiatry, Li Ka Shing Faculty of Medicine, The University of Hong Kong, Hong Kong, China; kfchung@hku.hk; 7Department of Psychology, The Chinese University of Hong Kong, Hong Kong, China; fionahoyy@cuhk.edu.hk

**Keywords:** insomnia, pandemic, cohort, mental health, web-based

## Abstract

In response to the worsening situation of the COVID pandemic, this follow-up study aimed to assess the impact of the “third wave” of the outbreak on sleep and mood disturbances among Hong Kong citizens. A total of 339 respondents included in our last study during the second wave (4–11 August 2020) joined this survey (response rate = 51.1%). The questionnaire collected data on sleep conditions, mood, stress, and risk perception. The sleep quality and mood status were assessed using the Insomnia Severity Index (ISI), General Anxiety Disorder-7 (GAD-7), and Patient Health Questionnaire-9. The weighted prevalence of insomnia, anxiety, and depression was 33.6%, 15.3%, and 22.0%, respectively. Compared with the last survey, five out of six sleep parameters significantly worsened despite the lack of difference in the ISI score. The GAD-7 score was significantly lower. Old-aged adults were less likely to maintain good sleep quality compared with middle-aged adults (adjusted OR = 4.03, 95% CI: 1.04–15.73). Respondents without psychiatric disorder were more likely to be anxiety-free across the two time points (adjusted OR = 7.12, 95% CI: 1.33–38.03). One-third of Hong Kong people reported poor sleeping quality in the third wave of the COVID-19 outbreak. Policy-makers need to propose a contingent plan to allocate mental health resources to vulnerable subpopulations.

## 1. Introduction

The impact of Coronavirus disease 2019 (COVID-19) on mental health is significant. A meta-analysis including 55 studies in multiple countries (*N* = 189,159) revealed that the pooled prevalence rates of insomnia, depression, anxiety, posttraumatic stress disorder, and psychological distress were 23.87%, 15.97%, 15.15%, 21.94%, and 13.29%, respectively. These prevalence estimates suggested that insomnia was the most prevalent symptom among the psychological sequelae [1]. These cross-sectional point-estimates provided timely insights into the psychological consequences of COVID-19, but the pandemic represented a highly dynamic occasion, which may be accompanied with changing severity of psychological sequelae. Therefore, longitudinal research is pivotal to enhance our understanding of the dynamics of psychological sequelae throughout the pandemic and corresponding risk and protective factors. Several large-scale longitudinal studies (sample sizes ranged from 2376 to 14,393) have been conducted to examine the changes in depressive and anxiety symptoms, on a time interval ranging from 2 weeks to 2 months [2,3,4,5], but mixed results were found. One study showed significant increase in mental disturbance [2], whereas other studies showed a slight increase or decrease in mental disturbance [3,4,5]. Younger age, females, high income, high education level, consumption of information about COVID-19, and low physical activity were risk factors associated with worsening psychological disturbance [2,5]. While researchers largely focused on the impact of pandemic on depression and anxiety, few studies assessed sleep disturbances, despite that sleep disturbance was listed as the most prevalent problem during the pandemic era [1]. 

Existing research examining the impact of COVID-19 on sleep was conducted using a cross-sectional survey, embedded with some posting questions retrospectively to draw inferences about participants’ change in sleep from one time point to another (e.g., before and after lock down) [6,7]. However, findings collected by retrospective methods were prone to recall bias. To our knowledge, only two small-scale research studies with repeated sleep measurements were conducted during the COVID-19 pandemic period. A study on 75 Spanish university students revealed that the prevalence of sleep disorders increased from 37.3% to 70.7% after 20 days of confinement [8]. In contrast, another study in Italy demonstrated a decrease of sleep problems among 37 preschool children throughout the 30-day study period [9]. Findings that emerged from retrospective studies have yielded inconsistent results. Hence, more longitudinal studies investigating the change of sleep quality are warranted. 

Hong Kong is one of the most densely populated regions in the world. It is an urban city with the vast majority of the population being ethnic Chinese. Similar to some other countries/regions, Hong Kong has experienced several COVID-19 epidemic waves in 2020. During “the second wave” (March–May 2020), the number of the confirmed and probable cases first reached 1000 in Hong Kong [10]. The situation seemed to improve from mid-April to June 2020 as there were periods of consecutive days without local transmission cases. However, in early July 2020, there was a re-emergence of locally acquired cases in Hong Kong, which indicated “the third wave” of COVID-19 pandemic [11]. The situation kept worsening, with over a hundred new cases per day reported in mid-July for the first time. The number of cases surged from 1233 before July to over 4000 in early August while fatal cases increased fourfold [10]. While the severity of outbreak raised, the sleep and mood disturbances of people in Hong Kong may deteriorate.

We conducted a cross-sectional survey involving 1138 Hong Kong respondents in early April 2020, which was the time when Hong Kong was just hit by “the second wave” of the outbreak [12]. A high prevalence of insomnia 29.9% was found. Insufficient mask supply was significantly associated with sleep quality (adjusted OR = 1.96). Since April 2020, Hong Kong has gone from no local manufacturing to over a hundred factories producing masks with support from the government. With the increasing abundance of mask supplies, the influence of the insufficient stock of masks on insomnia was likely to be limited. In response to the rising number of confirmed cases in the “third wave”, whether other factors may influence the sleep quality of the general public remained unknown. While the severity of outbreak raised, the sleep and mood disturbances of people in Hong Kong may deteriorate. Thus, this follow-up study aimed to (1) assess the impact of the “third wave” of the COVID-19 outbreak on the severity of sleep and mood disturbances among Hong Kong citizens, (2) compare the impact of the third wave with the second wave of the outbreak on sleep and mood, and (3) explore the factors associated with changes of sleep and mood impairments across two waves of the outbreak.

## 2. Materials and Methods

### 2.1. Study Design and Participants

Our previous survey was conducted during the second wave outbreak of COVID-19 in Hong Kong from 6 to 20 April 2020 (Time point 1). Respondents were asked to provide their contact details if they were willing to be contacted for future research at a later time. A follow-up survey was delivered to respondents via e-mail or WhatsApp using an online platform from 4 to 11 August 2020 (Time point 2). No additional inclusion or exclusion criteria was applied when recruiting participants. Over a 7-day survey period, 663 subjects who had left their contact details were contacted, of which 339 respondents returned the questionnaire (response rate: 51.1%).

### 2.2. Questionnaire

The follow-up questionnaire covered sleep conditions, mood and stress, and risk perception [12] as well as respondents’ exposure to confirmed cases of COVID-19, history of chronic diseases, and mental illnesses. The same set of psychological instruments were used to assess the insomnia symptoms and daytime impairment in the previous 2 weeks [12]. These include the Chinese version of the Insomnia Severity Index (ISI) with a cut-off value of 10 points (sensitivity: 86.1%; specificity: 87.7%) [13,14]. Sleep-wake parameters, including their sleep onset latency (SOL), wake time after sleep onset (WASO), early morning awakening (EMA), duration of insomnia symptoms if any, total sleep time (TST), and usual bed and rise times were obtained by the Chinese version of the Brief Insomnia Questionnaire (BIQ) [15,16]. Sleep efficiency (SE) was then determined (TST/total time in bed × 100%). The half-way point of sleep (mid-sleep point) was estimated by bed time, rise time, and SOL. The Chinese version of General Anxiety Disorder-7 (GAD-7) consists of seven items with an established cut-off point of 10 (sensitivity: 89%; specificity: 82%) that were used to assess anxiety [17,18]. Patient Health Questionnaire-9 (PHQ-9) is a nine-item scale that assesses depression. It has a similar cut-off point of 10 marks, which indicates major depression (sensitivity and specificity: 88%) [19,20]. 

Respondents’ worries about being infected and about their family members being infected, as well as their levels of stress and mood during the previous 7 days, were re-assessed in this follow-up survey using a 5-point Likert-like scale. Their risk perception of being infected, confidence in the ability to protect themselves and families from getting infected, and confidence in the ability of health professionals and the government to combat COVID-19 were re-evaluated.

### 2.3. Data Analysis

The SPSS 26.0 (SPSS Inc., Chicago, IL, USA) was used to analyze the data. Sociodemographic data, use of social media, and the sources of COVID-19 information were collected in the previous survey, which were summarized using descriptive statistics weighted with the 2016 Hong Kong population census distribution by age and gender. 

The changes in the sleep quality and mood status of the respondents were the primary outcome of this follow-up survey. Results of this follow-up survey were compared with data obtained in our last survey, with the exception of PHQ-9 score, which was not assessed previously. The ISI, GAD-7, and PHQ-9 scores of the respondents were coded into binary data according to the recommended clinical cut-off point for insomnia, anxiety, and depression. Based on the status of insomnia in the two surveys, the changes between the two time points were categorized into “deterioration, maintenance, remission, and persistent”. “Deterioration” referred to those having no clinical insomnia (ISI score < 10) at time point 1, but having clinical insomnia (ISI score ≥ 10) at time point 2; “maintenance” referred to those having no clinical insomnia in both time points; “remission” referred to those having clinical insomnia at time point 1, but having no clinical insomnia at time point 2; and “persistent” referred to those having clinical insomnia at both time points. The same categorization was applied to anxiety using GAD-7 score using the cut off <10 vs. ≥10. Changes of sleep condition, mood status, and risk perception between the two time points were compared using paired t-test or McNemar’s test.

Regression analyses were performed to explore the association of predictive factors with changes in the status of insomnia and anxiety across the two time points, whereas the associations of factors with changes in the ISI and GAD-7 scores were analyzed using Pearson correlation tests. The use of univariate regression analyses and Pearson correlations were used to describe the distribution of predictive factors among different sleep and mood status as a first step. Variables including age, gender, educational attainment, marital status, employment status, sources of acquiring COVID-19 information, social media use, worries about oneself and one’s family being infected, interference in daily life, stress during the pandemic, presence of chronic disease or psychiatric disorders, as well as confidence in self and family protection, ability of health professionals and the government to combat COVID-19, were predesignated and included in all the univariate analysis models. Besides, baseline ISI scores were included in the models evaluating mood while GAD-7 scores were used in the models evaluating sleep. Variables that were significant predictors (*p* < 0.10) in the univariate and correlation analysis were entered in the multivariate analysis to explore significant associations [21]. The inclusion level was set to 0.10 to improve the chances of retaining meaningful confounders [21]. The identified covariates were presented in terms of Odds Ratios (ORs) and 95% confidence intervals or correlation coefficient (r or R2) and standardized regression coefficient (β).

## 3. Results

### 3.1. Characteristics for the Respondents

In general, the 339 respondents’ sociodemographic characteristics included in the final sample were comparable with those non-completers of the follow-up survey (all *p* > 0.05), except the lower proportion of older population (aged above 40 years; *p* = 0.003) and a higher percentage of participants who were economically inactive (*p* = 0.04) (Table 1). Most respondents of the present study were female (*n* = 228, 67.3%) and young adults between 18 and 39 (58.7%). Half of them were married (50.7%). A vast majority of respondents (92%) were living with family members. Social media (67.0%) and conventional media (61.9%) were to be the most common sources of COVID-19-related information. Over half of our respondents (55%) spent more than 2 h on social media daily. 

Concerning exposure of COVID-19, 84.4% (*n* = 286) participants reported no exposure to any confirmed case. Among those who reported previous exposure to confirmed cases (*n* = 53, 15.6%), friends/colleagues (*n* = 26, 7.7%) or neighbors (*n* = 19, 5.6%) were the most common source of contact. Only two respondents (0.5%) were exposed to family members who were confirmed COVID-19 cases. 

### 3.2. Sleep Quality, Mood Status, and Risk Perceptions of the Respondents

Table 2 shows the findings on sleep, mood status, and risk perceptions of the respondents in the two time points. The mean SOL, WASO, and EMA of the respondents were 26.9, 31.8, and 36.6 min, respectively. The mean TST was 6.6 h with a mid-sleep point of 4:19 am, whereas the mean SE of respondents was 82.0%. The mean ISI score was 7.7 and 114 respondents (weighted prevalence: 33.6%) were classified as having clinical insomnia (ISI ≥ 10).

Significant differences were found in the five out of six sleep parameters between two time points. The means of WASO and EMA among the respondents increased by 8.7 and 15.1 min, respectively, whereas the mid-sleep point was 14 min earlier. The average TST decreased by 0.2 h, whereas average SE decreased by 4.5% (All *p* < 0.05). Nevertheless, no significant difference was found in the ISI score and the proportion of having clinical insomnia (ISI score ≥ 10) between the two time points. Of the 114 respondents who were classified as having clinical insomnia, 76 (67.0%) of them had persistent insomnia in both time points, whereas 38 (33.0%) had their sleep deteriorated. The sleep conditions of 37 respondents (10.9%) improved across the two time points.

Regarding mood status, the GAD-7 score was lower than that of the time point 1 (4.8 at time point 2 vs 5.2 at time point 1, *p* = 0.047). The proportion of the respondents (15.3%) suffering from clinical anxiety (GAD-7 ≥ 10) did not differ significantly from the time point 1 (18.3%, *p* = 0.22). Comparing the two time points, 32 (9.5%) of the respondents were classified as having anxiety in both surveys, whereas 20 (5.9%) were only classified as anxiety in this follow-up survey. Remission of clinical anxiety was reported in 30 (8.9%) of the respondents. For depression, the mean PHQ-9 score was 5.8 and 75 (22%) of the respondents were graded as clinical depression. 

Around 30% of respondents (*n* = 99) experienced a high level of financial stress, which was significantly lower than that in our previous survey (40.3%, *p* < 0.001). Notably, fewer respondents reported a lower level of confidence in self-protection (57.6% vs. 64.4%, *p* = 0.044), whereas no significant difference was found in their level of confidence on the ability of health professionals and the government to combat COVID-19.

### 3.3. Factors Associated with Changes in Sleep Quality across the Two Time Points

Multivariate analysis revealed that age and GAD-7 score were significantly associated with the sleep quality (*p* < 0.05, Table 3). Middle-aged adults (40–59 years) were four times more likely than the elderly (60 years or above) to maintain good sleep quality across the two time points (adjusted OR = 4.03, 95% CI [1.04–15.73], *p* = 0.045). Increasing anxiety score was associated with reduced odds of maintaining good sleeping quality (adjusted OR = 0.89, 95% CI [0.80–0.99], *p* = 0.04). 

The regression model found that (F = 10.86, adjusted R2 = 0.13, *p* < 0.001) a higher education level (*r* = −0.19) and GAD-7 score (*r* = −0.24) were negatively correlated with the changes in ISI score. Perceived higher economic stress (*r* = 0.15) was positively correlated with changes in ISI score (all *p* < 0.05) (Table 4).

### 3.4. Factors Associated with Changes in Anxiety across the Two Time Points

Multivariate regression model showed that increasing ISI scores was negatively associated with anxiety (adjusted OR = 0.84, 95% CI [0.75–0.93], *p* = 0.001, Table 3). Respondents without previously diagnosed psychiatric disorder were seven times more likely to be anxiety-free across these two time points (adjusted OR = 7.12, 95% CI [1.33–38.03], *p* = 0.02). In addition, regression model found that (F = 6.52, adjusted R2 = 0.12, *p* < 0.001) a higher stress level (*r* = −0.32) was negatively correlated with the changes in GAD-7 score (*p* < 0.001, Table 4).

## 4. Discussion

To the best of our knowledge, this was the first follow-up study on the prevalence of insomnia among the urban Chinese population across a long time-period during the COVID-19 pandemic. The overall prevalence of insomnia was similar in the second and third waves of the outbreak (33.4% vs. 33.6%), implying that sleep problem was common among Hong Kong citizens during the pandemic. A previous local survey in 2014 found that 32.7% of the population had insomnia symptoms at least three night per week accompanied with daytime impairment or distress [22], which was similar to the current findings. The prevalence rates of anxiety and depression during the third wave of the outbreak were 15.3% and 22.0%, respectively. Old age and higher GAD-7 score at time point 1 were associated with insomnia or worsening in sleep quality, whereas the history of psychiatric disorder and high ISI score at time point 1 were associated with worsening anxiety level. Although the number of COVID cases increased geometrically, no significant difference was found in the risk perception of self and family members being infected, whereas fewer respondents had a lower level of confidence in self-protection.

While the average ISI score and number of respondents who suffered from insomnia did not differ significantly across the two surveys, most of the sleep parameters deteriorated. This result implied that a reduction in sleeping quality occurred among the general population in the third wave of the outbreak compared with the second wave, although some of them might not be sufficiently classified as clinical insomnia. Increased information availability and knowledge on COVID-19 during the latter phase of the pandemic did not improve the sleeping quality among Hong Kong citizens. Older age and a high level of anxiety were associated with consistent insomnia during the pandemic. In fact, the elderly population was at higher risk of developing insomnia than other age groups, which could be attributed to multiple factors, such as circadian rhythm changes and psychosocial factors [23,24]. The elderly, especially those with cognitive decline, may be more anxious and agitated and become more vulnerable to developing mental disorders during the pandemic when they had to comply with social distancing to protect themselves from infection, thereby leading to social isolation/perceived loneliness [25]. Our claim was echoed by a local study, which reported that a significant increase in loneliness was found among older adults after the onset of the COVID-19 outbreak [26]. Many elderly relied on daycare/community centers and social services to maintain regular social contact, but these were not available during the pandemic. Prolonged social isolation may lead to increased loneliness and reduced connectedness. Additional attention should be paid to the older adults as they were more vulnerable to developing various mental disorders [27].

Despite the increasing number of newly confirmed COVID-19 cases in the third wave, our respondents had a generally higher level of confidence in self-protection compared with the preceding wave. The shortage of face masks was significantly associated with sleep disturbance in our previous study. Since the local outbreak, the local supply of masks had been stabilized. Two online surveys revealed that over 90% of local citizens considered mask wearing as a means to reduce the transmission of infection [28,29]. Therefore, the stable mask supply possibly boosted the sense of security among Hong Kong citizens. Moreover, mask-wearing offered relief from anxiety in another study [30] and high self-efficacy to wear facemask has better mental status [29] which may partly explain the reduced level of anxiety among the respondents in this follow-up study [27]. 

Although respondents’ general level of anxiety was reduced, this study reported that people with a history of psychiatric disorders were much more likely to suffer from anxiety during the pandemic. COVID-19 triggered increased fear, and the intense emotional responses to this fear may lead to mental relapse or aggravation of pre-existing mental health conditions [31]. People with psychiatric illnesses reportedly suffered from higher severity of stress, anxiety, insomnia, and depression, and they were more worried about their physical health than those without existing mental problems [32]. Additionally, the quarantine policy may particularly affect this subpopulation. During the surge of COVID-19 cases, people were less likely to regularly visit psychiatrists in out-patient clinics that could severely undermine respondents’ supervision and management of their psychiatric conditions. Access to care facilities was also prohibited during the pandemic, and thus, these patients were left at home for a long duration without any mental health support. 

This study had several limitations. First, among the 663 participants who were contacted for future research, only 51.9% were assessed at this follow-up survey. Nevertheless, most of the socio-demographic characteristics of our respondents were comparable with our first survey. This ensured a valid comparison between the results of the two surveys in the same cohort. Second, the results of this survey were only based on participants who agreed to participate in the follow-up and have completed the survey. The prevalence of insomnia may be over-estimated/under-estimated due to attrition in the follow-up survey as people who suffered from insomnia may be less likely to participate for a long period of study [33]. Third, between the second and third waves of the COVID outbreak, there were fluctuations in the number of cases and the extent of spread. As we only captured the sleep changes between April and August 2020, we could not draw a comparison to identify the potential changes in the sleep patterns and quality between the two waves. Fourth, as this study was carried out among a Chinese population in Hong Kong, the finding may be specific to the population and may vary in samples from different geographic areas and environments. Finally, there may be other factors that may contribute to the change in sleep and mood disorders which have not been included in the present study such as vaccination status. Future studies can look at factors, such as vaccination status and compulsory testing on the sleep quality and mood of Hong Kong citizens.

## 5. Conclusions

About one-third of Hong Kong people still reported insomnia during the second wave of outbreak, and over 60% of them had their sleep problems to be persistent at the third wave of outbreak. Anxiety and depression were also reported in 15–22% of the respondents. Older age adults and those with existing psychiatric disorders were especially susceptible to sleep and mental health problems. Policy-makers need to allocate timely mental health resource and support to those in this vulnerable subpopulation.

## Figures and Tables

**Table 1 ijerph-18-08444-t001:** Characteristics of the respondents.

Variable	Subjects Only Contribute at Second Wave *N* = 799 (%)	Subjects Followed at Third Wave *N* = 339 (%)	*p*-Value ^b^
Female	519 (65.0)	228 (67.3)	0.46
Age			0.003
18–39 years	427 (53.4)	199 (58.7)	
40–59 years	327 (40.9)	108 (31.9)	
60 years or above	45 (5.6)	32 (9.4)	
Married (versus Single/Separated/Divorced)	389 (48.7)	170 (50.1)	0.65
No Child	606 (75.8)	264 (77.9)	0.46
Living with family members	743 (93.0)	312 (92.0)	0.57
Educational attainment			0.17
Secondary school (junior form) or below	27 (3.4)	7 (2.1)	
Secondary school (senior form)	157 (19.6)	55 (16.2)	
Tertiary education	615 (77.0)	277 (81.7)	
Employment status			0.04
Employed	617 (77.2)	241 (71.1)	
Unemployed	31 (3.9)	11 (3.2)	
Economically inactive	151 (18.9)	87 (25.7)	
With chronic disease	-	55 (16.2)	N/A
Time spent on social media per day			0.36
Non-user	2 (0.3)	2 (0.6)	
2 h or less	326 (40.8)	150 (44.2)	
More than 2 h	471 (58.9)	187 (55.2)	
Frequent source(s) of COVID-19 relevant information			
Governmental press conference	215 (26.9)	91 (26.8)	0.98
Health organization (e.g., World Health Organization and Centers for Disease Control and Prevention)	105 (13.1)	51 (15.0)	0.39
Press conference hosted by health professionals	184 (23.0)	87 (25.7)	0.34
Information disseminated at workplace	154 (19.3)	61 (18.0)	0.61
Conventional media (e.g., Newspapers)	485 (60.7)	210 (61.9)	0.69
Social media	503 (63.0)	227 (67.0)	0.20
COVID-19 exposure of confirmed cases			
Family member(s)	-	2 (0.5)	N/A
Friends/Colleague(s)	-	26 (7.7)	N/A
Neighbor(s)	-	19 (5.6)	N/A
Others ^a^	-	9 (2.7)	N/A
No exposure	-	286 (84.4)	N/A

Abbreviations: COVID-19, Coronavirus Disease 2019. ^a^ Exposure to confirmed cases, which are family members of colleagues, friends of friends, confirmed cases in the same district, cleaning workers in the same office/residential building, and confirmed cases that have visited offices or residential buildings. ^b^ Comparisons made using Chi-square test.

**Table 2 ijerph-18-08444-t002:** Sleep, mood status, and risk perceptions of the respondents (total sample = 339).

	Second-Wave ^a^	Third-Wave ^a^	*p*-Value ^b^
*Sleep Condition since COVID-19 Outbreak*			
Sleep Parameters			
SOL, minutes (*N* = 337)	26.9 ± 33.8	26.9 ± 28.6	0.97
WASO, minutes (*N* = 336)	23.1 ± 41.3	31.8 ± 55.3	0.004
EMA, minutes (*N* = 335)	21.5 ± 39.2	36.6 ± 51.2	<0.001
TST, hours (*N* = 338)	6.8 ± 1.3	6.6 ± 1.3	0.03
SE, % (*N* = 278)	86.5 ± 14.3	82.0 ± 15.1	<0.001
Mid-sleep point	4:33 ± 1:24	4:19 ± 1:19	<0.001
ISI, ranged 0–28	7.2 ± 5.2	7.7 ± 5.1	0.07
Clinical insomnia (ISI ≥ 10)	113 (33.4)	114 (33.6)	0.83
Changes between two time points			
Sleep deteriorated	/	38 (11.1)	/
Maintained good sleep	/	188 (55.5)	/
Remission of insomnia	/	37 (10.9)	/
Persistent insomnia	/	76 (22.5)	/
*Mood Status since COVID-19 Outbreak*			
GAD-7, ranged 0–21	5.2 ± 5.0	4.8 ± 4.7	0.047
Clinical anxiety (GAD-7 ≥ 10)	62 (18.3)	52 (15.3)	0.22
Changes between two time points (anxiety)			
Mood deteriorated	/	20 (5.9)	/
Maintain emotional health	/	257 (75.8)	/
Remission of anxiety	/	30 (8.9)	/
Persistent anxiety	/	32 (9.5)	/
PHQ-9, ranged 0–21	/	5.8 ± 5.2	/
Clinical depression	/	75 (22.0)	/
Interfered with daily life due to COVID-19 (much, very much)	264 (77.9)	271 (80.1)	0.40
Experiencing stress (much, very much)	146 (43.2)	133 (39.4)	0.15
Experiencing Financial stress (much, very much)	137 (40.3)	99 (29.1)	<0.001
*Risk Perceptions since COVID-19 Outbreak*			
Worrying about being infected (Somewhat, Much, or Very much)			
Own self	171 (50.5)	141 (41.7)	<0.001
Family members	182 (53.8)	175 (51.7)	0.10
Confidence against COVID-19			
Self-protection	218 (64.4)	195 (57.6)	0.044
Health professional	282 (83.1)	284 (83.8)	1.00
The government	111 (32.7)	97 (28.7)	0.14

Abbreviations: COVID-19, Coronavirus Disease 2019; EMA, early morning awakening; GAD-7, the 7-item General Anxiety Disorder; ISI, Insomnia Severity Index; SE, sleep efficiency; SOL, sleep onset latency; TST, total sleep time; WASO, wake after sleep onset. ^a^ Data presented in Number (Weighted Percentage) or Weighted Mean ± Standard Deviation. ^b^ Comparison made based on paired t-test or McNemar’s test.

**Table 3 ijerph-18-08444-t003:** Factors associated with worsening or persistent mood and sleep problems by univariate and multivariate logistic regression.

Variables	Univariate		Multivariate	
	OR	95% CI	OR	95% CI
**Maintenance of good sleep (versus sleep deterioration) *N* = 226**				
Age				
Young adults (versus elderly)	3.56 *	1.23–10.29	2.63	0.71–9.79
Middle-aged adults (versus elderly)	3.76 *	1.20–11.79	4.03 *	1.04–15.73
Tertiary education (versus primary/secondary schooling)	2.33 *	1.04–5.22	2.58	0.96–6.92
Employment				
Unemployed (versus economically inactive)	0.19 *	0.04–0.95	0.27	0.04–1.84
Employed (versus economically inactive)	1.07	0.45–2.55	078	0.30–2.08
GAD-7 score	0.89) *	0.81–0.98	0.89 *	0.80–0.99
Not having distress	1.83	0.90–3.70	1.33	0.56–3.12
Without chronic disease	2.44 *	1.05–5.65	1.70	0.60–4.80
Without psychiatric disorder	3.94	0.85–18.39	3.61	0.64–20.30
**Maintenance of Emotional Health (Anxiety-free) (versus Mood Deterioration) *N* = 276**				
ISI score	0.82 ***	0.74–0.91	0.84 **	0.75–0.93
Not having economic stress	2.67 *	1.07–6.69	2.01	0.76–5.32
Without psychiatric disorder	6.92 *	1.60–29.97	7.12 *	1.33–38.03

Abbreviations: 95% CI, Confidence Interval; GAD-7, the 7-item General Anxiety Disorder; ISI, Insomnia Severity Index; OR, Odds Ratio; * *p* < 0.05, ** *p* < 0.01, *** *p* < 0.001.

**Table 4 ijerph-18-08444-t004:** Predictors of sleep and mood changes from the second- to third-wave outbreak of COVID-19.

Variables	*r* ^a^	Standardized β	t	Adjusted R^2^	F
**Sleep changes (ISI score)**					
Age	0.12	0.05	0.87	0.13	10.86 ***
Education level	−0.19	−0.15	−2.77 **		
Stress level	−0.19	−0.05	−0.80		
Perception of economic stress	0.15	0.22	4.05 ***		
GAD-7 score	−0.24	−0.26	−4.03 ***		
**Mood changes (GAD-7 score)**					
Education level	−0.11	−0.09	−1.72	0.12	6.52 ***
The risk perception about being infected	−0.18	−0.09	−1.30		
The risk perception about family members being infected	−0.17	0.03	0.38		
Confidence in self-protection	0.10	0.03	0.53		
Confidence in health professionals	0.10	0.10	1.79		
The perceived interference of daily living	−0.10	0.06	1.01		
Stress level	−0.32	−0.27	−4.42 ***		
ISI score	−0.22	−0.10	−1.66		

Abbreviations: GAD-7, the 7-item General Anxiety Disorder; ISI, Insomnia Severity Index. ^a^ Factors with a *p*-value less than 0.1 in the Pearson correlation were included in the multiple linear regression model. ** *p* < 0.01, *** *p* < 0.001.

## Data Availability

Not applicable.

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
