# Peer review of "Sleep and Mood Disturbances during the COVID-19 Outbreak in an Urban Chinese Population in Hong Kong: A Longitudinal Study of the Second and Third Waves of the Outbreak"

_ijerph, 2021, doi:10.3390/ijerph18168444_

Round 1
Reviewer 1 Report
Major comments
- The statistical methods section on Line 133. It would be good if the authors could explain why a cutoff of p <0.1 was selected as indicating “significance”. Also, when there are many variables tested in univariate analysis, multiple comparison should be considered and p value needs to be further adjusted to control for false positive findings.
- Line 134, The variables selected into the multivariate analysis should be based on the hypothesized relationship between different factors and the predictor-outcome relationships, rather than using statistics to select them. It would also be good to make it clear about what variables were selected in each model.
- Some of the discussion is confusing, e.g. Line 240, I don’t find results indicate that repsondents in the third wave had a generally higher level of confidence in self-protection, as it is not significant in the multivariate table, and the direction is reversed in the univariate table.
Minor comments:
- It would be beneficial to present the prevalence of the outcomes before pandemic in Hong Kong, to demonstrate the negative impacts of the pandemic.
- Line 140, Table 1 is not clear regarding how the characteristics were comparable between previous survey and this survey.
Author Response
We thank the Reviewer for giving us valuable suggestions and allowing us to revise the manuscript. We have responded to the comments as follow:
- The statistical methods section on Line 133. It would be good if the authors could explain why a cutoff of p <0.1 was selected as indicating “significance”. Also, when there are many variables tested in univariate analysis, multiple comparison should be considered and p value needs to be further adjusted to control for false positive findings.
Re: Thank you for the comment. The cutoff of p<0.1 was based on the study by Bursac et al. Their study showed that the purposeful selection algorithm (which begins by univariate analyses of each variable) identifies and retains confounders correctly at a larger rate particularly in instances where the significance level is between 0.1 and 0.15. More traditional levels such as 0.05 can fail in identifying variables known to be important.
Ref: Bursac Z, Gauss CH, Williams DK, Hosmer DW. Purposeful selection of variables in logistic regression. Source code for biology and medicine. 2008 Dec;3(1):1-8.
https://www.ncbi.nlm.nih.gov/pmc/articles/PMC2633005/
2. Line 134, The variables selected into the multivariate analysis should be based on the hypothesized relationship between different factors and the predictor-outcome relationships, rather than using statistics to select them. It would also be good to make it clear about what variables were selected in each model.
RE: Thank you for the comments. We pre-specified a list of covariates and socio-demographic factors that should be examined in univariate analysis according to the hypothesized relationship between these covariates and sleep/mood outcomes. For the multivariate analysis, we then purposefully narrowed down to only include significant predictors of sleep/mood outcome (as p < 0.1) as suggested by Bursac et al. (2008).
- Some of the discussion is confusing, e.g. Line 240, I don’t find results indicate that repsondents in the third wave had a generally higher level of confidence in self-protection, as it is not significant in the multivariate table, and the direction is reversed in the univariate table.
Re: Thank you letting us clarify, this discussion is based on the finding that “Notably, fewer respondents reported a lower level of confidence in self-protection (57.6% vs 64.4%, p = 0.044)” (Results section 3.2, Line 201-203 & Table 2), which is a result based on the total sample of respondents.
Minor comments:
- It would be beneficial to present the prevalence of the outcomes before pandemic in Hong Kong, to demonstrate the negative impacts of the pandemic.
Re: Thank you for the suggestion. We have added the statistics on the prevalence of insomnia before pandemic as comparison A previous local survey in 2014 found that 32.7% of the population had insomnia symptoms at least three night per week accompanied with daytime impairment or distress (Chung et al., 2015), which was similar to the current findings (Discussion, Line 234-236).
Ref: Chung KF, Yeung WF, Ho FY, Yung KP, Yu YM, Kwok CW. Cross-cultural and comparative epidemiology of insomnia: the Diagnostic and statistical manual (DSM), International classification of diseases (ICD) and International classification of sleep disorders (ICSD). Sleep Med. 2015 Apr;16(4):477-82. doi: 10.1016/j.sleep.2014.10.018
2. Line 140, Table 1 is not clear regarding how the characteristics were comparable between previous survey and this survey.
Response: Thank you for the useful suggestion. We added Chi-square test for demonstrating the comparability of subjects’ characteristics between completers and non-completers of follow-up survey at third wave (Table 1, Results 3.1). Compared with respondents only contributed data at the second epidemic wave, the follow-up sample only differed in slightly lower percentage of middle-aged and elderly and the increased proportion of those who were economically inactive (p < 0.05).
We hope the Reviewer now finds the revised manuscript is acceptable for publication.
Reviewer 2 Report
The objective that the authors declared in this manuscript was to assess the impact of the “third wave” of the outbreak on sleep and mood disturbances among Hong Kong citizens. The main findings were a persistence of insomnia in third wave and older age adults and those with existing psychiatric disorders were espe cially susceptible to sleep and mental health problems.
There are currently a large number of reports including some meta-analyzes and systematic reviews in this regard, but as pointed out by the authors, there are few studies that have followed up on these problems.
Observations
1. The objectives of the study are not congruent with the title, since the latter does not speak of the "waves" but in general of the outbreak
2. An important characteristic of this study is that both, first measurement and in the second evaluation the data were collected during a “wave”, however, waves can vary significantly between them, for example with respect to their duration and number of people affected, the present study does not provide any information on the general characteristics of each wave, in particular the differences between the second and third waves.
3. There is no hypothesis of the differences between the second and third waves that were expected to be found, and due to the lack of information on the waves referred to above, it is not possible to infer it (Were sleep and mood disturbances expected to decrease or increase?)
4. I consider that the main weakness of this study is the lack of information from the participants. In the study previously published by the authors, only a few selection criteria are mentioned, but the sample size is greater than that reported in the present study as "second wave", Which was the criterion to consider these participants as" second wave "since it is mentioned that it was carried out in the" first phase "?. In the present study, apparently the only selection criterion was the fact of having their contact details and responded to the second interview, it should be noted if there were no elimination criteria. It is also not clear whether covid 19 infection was an exclusion criterion since nothing is reported in this regard (acute or recovery infection) only possible sources of exposure are mentioned, another important bias could have been the presence of an acute disease or the exacerbation of a chronic disease or drug useat the time of the interview, among others.
5. Table 1 shows the demographic data only from the third wave, but the data from the second wave is not presented, it is only mentioned that it was comparable but the values ​​or statistical analyzes that confirm the statement are not shown.
6. The n presented in table 2 must be reported for each wave
7. In table 3, the absence of psychiatric disorder was found as a factor associated with Maintenance of Emotional Health. Which psychiatric disorders were considered?
8. In the discussion, the availability of masks is mentioned as a factor that can contribute to sleep and mood disorders. Why was this variable not analyzed in this study, while in the second wave it is reported as a risk factor? ? In the same way, other risk factors that may have changed from one wave to another could be analyzed, such as the time of confinement, the availability of vaccines or the vaccination status.
9. Given the large number of studies in this regard, the longitudinal results should be highlighted and differences between the two moments in which the evaluations were carried out should be discussed, since the fact that insomnia has not decreased significantly is interesting because it could be expected that with more resources and knowledge that people have about COVID 19 they could improve psychiatric symptoms
Author Response
We thank the Reviewer for giving us valuable suggestions and allowing us to revise the manuscript. We have responded to the comments as follow:
Observations
1. The objectives of the study are not congruent with the title, since the latter does not speak of the "waves" but in general of the outbreak
RE: Thanks for the suggestions. We revise the title to “ Sleep and mood disturbances during the COVID-19 outbreak in an urban Chinese population in Hong Kong: A longitudinal study of the ‘second’ and ‘third wave’ of the outbreak”
And Introduction (last paragraph):“Thus, this follow-up study aimed to 1) assess the impact of the “third wave” of the COVID-19 outbreak on the severity of sleep and mood disturbances among Hong Kong citizens, 2) compare the impact of the ‘third wave’ with the “second wave” of the outbreak and 3) explore the factors associated with changes of sleep and mood impairments across two waves of the outbreak (Introduction, Line 90-91).
- An important characteristic of this study is that both, first measurement and in the second evaluation the data were collected during a “wave”, however, waves can vary significantly between them, for example with respect to their duration and number of people affected, the present study does not provide any information on the general characteristics of each wave, in particular the differences between the second and third waves.
RE: Thank you for raising this important point. We have added descriptions on the second and third waves of the outbreak in Hong Kong in the introduction part.
Introduction (paragraph 3):
“Hong Kong is one of the most densely populated regions in the world. It is an urban city with the vast majority of the population being ethnic Chinese. Similar to some other countries/regions, Hong Kong has experienced several COVID-19 epidemic waves in 2020. During “the second wave” (Mar-May 2020), the number of the confirmed and probable cases first reached 1000 in Hong Kong. [1] The situation seemed to improve from Mid-April to June 2020 as there were periods of consecutive days without local transmission cases. However, in early July 2020, there was a re-emergence of locally acquired cases in Hong Kong, which indicated “the third wave” of COVID-19 pandemic. [2] The situation kept worsening with over a hundred new cases per day reported in mid-July for the first time. The number of cases surged from 1,233 before July to over 4,000 in early August while fatal cases increased fourfold. [1] While the severity of outbreak raised, the sleep and mood disturbances of people in Hong Kong may deteriorate.
Ref: [1] Centre of Health Protection DoH. Latest situations of cases of COVID-19. Hong Kong: Department of Health of the HKSAR. 2020.
[2] To, K. K. W., Chan, W. M., Ip, J. D., Chu, A. W. H., Tam, A. R., Liu, R., et al. (2020). Unique SARS-CoV-2 clusters causing a large COVID-19 outbreak in Hong Kong. Clinical Infectious Diseases. 2021 Jul 1;73(1):137-142.
- There is no hypothesis of the differences between the second and third waves that were expected to be found, and due to the lack of information on the waves referred to above, it is not possible to infer it (Were sleep and mood disturbances expected to decrease or increase?)
RE: Thank you for the comments. We have add the comparison between third wave and second wave of the outbreak on sleep and mood as an objective (Introduction, Line 90-91) and added our expectation that the sleep and mood disturbances of people in Hong Kong may deteriorate (Introduction, Line 87-88).
- I consider that the main weakness of this study is the lack of information from the participants. In the study previously published by the authors, only a few selection criteria are mentioned, but the sample size is greater than that reported in the present study as "second wave", Which was the criterion to consider these participants as" second wave "since it is mentioned that it was carried out in the" first phase "?. In the present study, apparently the only selection criterion was the fact of having their contact details and responded to the second interview, it should be noted if there were no elimination criteria. It is also not clear whether covid 19 infection was an exclusion criterion since nothing is reported in this regard (acute or recovery infection) only possible sources of exposure are mentioned, another important bias could have been the presence of an acute disease or the exacerbation of a chronic disease or drug useat the time of the interview, among others.
RE: Thank you for letting us clarify. We did not add extra exclusion criteria that differed from our previously study during the “second wave” of the outbreak when including participants in this follow-up survey to ensure comparability. This has been added into the method section. We did not exclude the those had infected and recovered from COVID-19 but the items regarding previous exposure of COVID-19 confirmed cases already included themselves being infected (2.1. Study design and participants, Line 99-100). However, none of the respondents reported acute or previous COVID-19 infection. The presence of chronic disease is already included in univariate analysis (being a potential predictor) of sleep and mood changes as suggested by reviewers. Nevertheless, the presence of chronic disease was not associated significantly with sleep/mood deterioration.
- Table 1 shows the demographic data only from the third wave, but the data from the second wave is not presented, it is only mentioned that it was comparable but the values ​​or statistical analyzes that confirm the statement are not shown.
RE: Thank you for the useful suggestion. We now included the characteristics of the respondents in Table 1.
- The n presented in table 2 must be reported for each wave
RE: Comparisons in sleep, mood state and risk perception were conducted only in 339 respondents with completed data (contribute data in both the second wave and third wave).
- In table 3, the absence of psychiatric disorder was found as a factor associated with Maintenance of Emotional Health. Which psychiatric disorders were considered?
RE: Yao et al (2020) suggested individuals with pre-existing mental conditions could be substantially impacted by the emotional burden of COVID-19 pandemic. Psychiatric disorders include previously diagnosed depression, bipolar disorder, schizophrenia and other psychoses, dementia, as well as developmental disorders including autism, as listed in the World Health Organization fact sheets. The scope has been added as a footnote in Table 3.
Reference:
Yao, H., Chen, J.-H., & Xu, Y.-F. (2020). Patients with mental health disorders in the COVID-19 epidemic. The Lancet Psychiatry, 7(4), e21. https://doi.org/10.1016/S2215-0366(20)30090-0
- In the discussion, the availability of masks is mentioned as a factor that can contribute to sleep and mood disorders. Why was this variable not analyzed in this study, while in the second wave it is reported as a risk factor? ? In the same way, other risk factors that may have changed from one wave to another could be analyzed, such as the time of confinement, the availability of vaccines or the vaccination status.
RE: Thank you for raising these important points. The mask supplies have been stabilized since our last study, therefore its impact on sleep quality and mood was not considered to be significant (Introduction, Line 81-84). For vaccination, vaccines were not available in Hong Kong until Feb 2021 and there had been no reported timeline during the time of study. Therefore we did not include this factor in the analysis.
We recognize there might be some factors that could contribute to change in sleep and mood disorders; however, we include the factors in our last study so that a comparison can be made. Nevertheless, we include this as a limitation and possible direction for future studies (Discussion, Line 302-304).
- Given the large number of studies in this regard, the longitudinal results should be highlighted and differences between the two moments in which the evaluations were carried out should be discussed, since the fact that insomnia has not decreased significantly is interesting because it could be expected that with more resources and knowledge that people have about COVID 19 they could improve psychiatric symptoms
RE: Thank you for the suggestion. We have already included in our discussion the longitudinal results, such as:
Discussion (paragraph 2):
“While the average ISI score and number of respondents who suffered from insomnia did not differ significantly across the two surveys, most of the sleep parameters deteriorated. This result implied that a reduction in sleeping quality occurred among the general population in the third wave of the outbreak compared with the second wave…”
“Despite the increasing number of newly confirmed COVID-19 cases in the third wave, our respondents had a generally higher level of confidence in self-protection compared with the preceding wave…”
“Although respondents’ general level of anxiety was reduced, this study reported that people with a history of psychiatric disorders were much more likely to suffer from anxiety during the pandemic...”
In addition, we have added some highlights of our longitudinal results:
“While the average ISI score and number of respondents who suffered from insomnia did not differ significantly across the two surveys, most of the sleep parameters deteriorated. This result implied that a reduction in sleeping quality occurred among the general population in the third wave of the outbreak compared with the second wave, although some of them might not be sufficiently classified as clinical insomnia. Increase information availability and knowledge on COVID-19 during the latter phase of the pandemic did not improve the sleeping quality among Hong Kong citizens…” (Discussion, Line 247-250)
We hope the Reviewer now finds the revised manuscript is acceptable for publication.
Reviewer 3 Report
Lam and collaborators aimed to investigate the impact of a third COVID-19 wave on sleep disturbances in a Hong Kong sample. The studied research topic is important for the mental health of the general population during the pandemic. All sections in the manuscript are clearly presented and the information is complete. I just have a few recommendations for the authors:
- I suggest using the term “major depression disorder” instead of depression in the abstract and the whole text, as the former is the correct name of the described disorder.
- In line 187, I suggest specifying what is the term “middle-aged” refers to in order to make this sentence clear without seeing the corresponding table
- In line 199: the term “psychiatric disorders” is referring to the psychiatric disorders evaluated in the present study or to those previously diagnosed?
- I suggest adding as a limitation that findings are potentially population-specific and are likely to vary in samples from different geographic areas and environments.
Author Response
We thank the Reviewer for giving us valuable suggestions and allowing us to revise the manuscript. We have responded to the comments as follow:
- I suggest using the term “major depression disorder” instead of depression in the abstract and the whole text, as the former is the correct name of the described disorder.
RE: Thanks for pointing out. However, after deliberation, we are afraid that it may not be appropriate to use the term “major depression disorder” since we only used PHQ cut-off instead of using the diagnostic criteria according to the DSM-5.
2. In line 187, I suggest specifying what is the term “middle-aged” refers to in order to make this sentence clear without seeing the corresponding table
Response: Thank you for letting us clarify. We have added the description to clarify the term “middle-aged”
“… Middle-aged adults (40-59 years) were four times more likely than the elderly (60 years or above) to maintain good sleep quality across the two time points (adjusted OR = 4.03, 95% CI [1.04–15.73], p = 0.045).” Results section 3.2 (Line 207-208):
3. In line 199: the term “psychiatric disorders” is referring to the psychiatric disorders evaluated in the present study or to those previously diagnosed?
Response: Thank you for letting us clarify. “Psychiatric disorder” refers to those previously diagnosed. We have made clarifications in the texts.
Results section 3.4 (Line 219):
“… Respondents without previously diagnosed psychiatric disorders were seven times more likely to be anxiety-free across this two time points (adjusted OR = 7.12, 95% CI [1.33-38.03], p = 0.02).”
4. I suggest adding as a limitation that findings are potentially population-specific and are likely to vary in samples from different geographic areas and environments.
Response: Thank you for the useful comment. This is an important limitation regarding the generalizability of the finding. The limitation has been added in the discussion part.
Discussion (last paragraph):
“...Fourth, as this study was carried out among a Chinese population in Hong Kong, the finding may be specific to the population and may likely vary in samples from different geographic areas and environments.” (Line 298-300)
We hope the Reviewer now finds the revised manuscript is acceptable for publication.
Round 2
Reviewer 1 Report
Thanks for addressing the comments. However, I still feel my major comments 1 and comment 2 are not well addressed, especially comment 1.
For the first comment, if you use univariate analysis only for the purpose of selecting covariates then p<0.1 is fine. However, if you plan to use univariate analysis to interpret the association between those variables with the outcome, you should adjust your p-value to account for the multiple comparisons issues here. For example, if you examine 100 variables with the outcome, p<0.1 will give you 10 significant findings no matter what they are.
For comment 2, I would like to know if the variables adjusted for each model are added and specified to the manuscript. Please make it very clear to the reader, for each outcome, each model, what covariates are included. If the same set of covariates are included for all models, please state it as well.
Author Response
Thanks for addressing the comments. However, I still feel my major comments 1 and comment 2 are not well addressed, especially comment 1.
Response: We thank again the Reviewer for letting us clarify and allowing us to revise the manuscript. We have responded to the comments as follow:
For the first comment, if you use univariate analysis only for the purpose of selecting covariates then p<0.1 is fine. However, if you plan to use univariate analysis to interpret the association between those variables with the outcome, you should adjust your p-value to account for the multiple comparisons issues here. For example, if you examine 100 variables with the outcome, p<0.1 will give you 10 significant findings no matter what they are.
Response: Thank you for raising this important point. We only use univariate analysis for the purpose of selecting covariates, while only the results from multivariate analysis were utilized to explore the significance of associations. Previous studies using multivariate logistic regression analyses first tested all the variables in the univariate and those variables significant at a critical level ranged from p < 0.10 to p <0.25 were included in further multivariate model. [1-4] Thanks for the opportunities of making clarifications. The current paper we set the critical level of identifying predictive variables as p < 0.10, as suggested by Bursac et al to improve the chances of retaining meaningful confounders [5]. This is to ensure the test will identify variables that, by themselves, are not significantly related to the outcome but make an important contribution in the presence of other variables. Therefore, we revise the Materials and Methods section for clearer presentations.
Ref:
[1] Demchuk AM, Morgenstern LB, Krieger DW, et al. (1999). Serum Glucose Level and Diabetes Predict Tissue Plasminogen Activator–Related Intracerebral Hemorrhage in Acute Ischemic Stroke. Stroke. 1999;30:34–39.
[2] Hajsheikholeslami F, Hatami M, Hadaegh F, et al. Association of educational status with cardiovascular disease: Teheran Lipid and Glucose Study. Int J Public Health. 2011;56:281–287.
[3] Kim SH, Youn CS, Kim HJ, Choi SP. Prognostic Value of Serum Albumin at Admission for Neurologic Outcome with Targeted Temperature Management after Cardiac Arrest. Emergency Medicine International. 2019; 6132542.
[4] Ito G, Kawakami K, Aoyama T, et al. Risk factors for severe neutropenia in pancreatic cancer patients treated with gemcitabine/nab-paclitaxel combination therapy. PLoS One. 2021;16(7):e0254726.
[5] Bursac Z, Gauss CH, Williams DK, Hosmer DW. Purposeful selection of variables in logistic regression. Source code for biology and medicine. 2008;3(1):1-8.
Materials and methods (2.3 Data analysis Paragraph 3)
“Univariate Regression analyses were performed to explore the association of predictive factors with changes in the status of insomnia and anxiety across the two time points, whereas the associations of factors with changes in the ISI and GAD-7 scores were analyzed using Pearson correlation tests. The use of univariate regression analyses and Pearson correlations were used to describe the distribution of predictive factors among different sleep and mood status as a first step. Variables that were significant predictors (p < 0.10) in the univariate and correlation analysis were entered in the multivariate analysis to explore significant associations [21]. The inclusion level was set to 0.10 to improve the chances of retaining meaningful confounders [21]. The identified covariates were presented in terms of Odds Ratios (ORs) and 95% confidence intervals or correlation coefficient (r or R2) and standardized regression coefficient (β).”
For comment 2, I would like to know if the variables adjusted for each model are added and specified to the manuscript. Please make it very clear to the reader, for each outcome, each model, what covariates are included. If the same set of covariates are included for all models, please state it as well.
Response: Thank you so much for the comments. We have now specified all the predesignated variables in the models in the method section. The variables were the same for all models except for baseline ISI scores (for evaluating mood/mood changes) and GAD-7 scores (for evaluating sleep quality).
Materials and methods (2.3 Data analysis Paragraph 3)
“Univariate Regression analyses were performed to explore the association of predictive factors with changes in the status of insomnia and anxiety across the two time points, whereas the associations of factors with changes in the ISI and GAD-7 scores were analyzed using Pearson correlation tests. Variables including age, gender, educational attainment, marital status, employment status, sources of acquiring COVID-19 information, social media use, worries about oneself and one's family being infected, interference in daily life, stress during the pandemic, presence of chronic disease or psychiatric disorders, as well as confidence in self and family protection, ability of health professionals and the government to combat COVID-19, were predesignated and included in all the univariate analysis models. Besides, baseline ISI scores were included in the models evaluating mood while GAD-7 scores were used in the models evaluating sleep…”
Reviewer 2 Report
I consider that the observations were adequately addressed, now there is greater clarity in the objectives and hypotheses, also the methodology and results are presented appropriately
Author Response
We thank for the Reviewer's positive comment.